# Breastfeeding Disparities between Multiples and Singletons by NICU Discharge

**DOI:** 10.3390/nu11092191

**Published:** 2019-09-12

**Authors:** Roser Porta, Eva Capdevila, Francesc Botet, Gemma Ginovart, Elisenda Moliner, Marta Nicolàs, Antonio Gutiérrez, Jaume Ponce-Taylor, Sergio Verd

**Affiliations:** 1Neonatal Unit, Dexeus University Hospital, 5 Sabino Arana st, 08028 Barcelona, Spain; 2Pediatric Unit, Department of Primary Care, Catalonia Health Authority, Balmes st, 08007 Barcelona, Spain; e.capde@gmail.com; 3Neonatal Unit, University Maternity Hospital, 5 Sabino Arana st. 08028 Barcelona, Spain; fbotet@clinic.cat; 4Neonatal Unit, Santa Creu i Sant Pau University Hospital, 87 mSant Quinti st. 08041 Barcelona, Spainemoliner@santpau.cat (E.M.); 5Neonatal Unit, Germans Trias i Pujol University Hospital, Canyet Road, 08916 Badalona, Spain; mnicolasl.germanstrias@gencat.cat; 6Department of Hematology, Son Espases University Hospital, IdISBa Balearic Medical Research Council. Valldemossa Road, 79, 07010 Palma de Mallorca, Spain; 7COMIB Advisory, Passeig de Mallorca, 42, 07012 Palma de Mallorca, Spain; 8Urgent Care Centre, Department of Primary Care, Balearic Health Authority, 1 Illes Balears st. 07014 Palma de Mallorca, Spain; Jaume.ponce@gmail.com; 9Pediatric Unit, Department of Primary Care, Balearic Health Authority, Matamusinos st. 07013 Palma de Mallorca, Spain

**Keywords:** breastfeeding, multiple pregnancy, neonate, premature birth, milk bank, pregnancy outcomes

## Abstract

Multiple pregnancy increases the risk of a range of adverse perinatal outcomes, including breastfeeding failure. However, studies on predictive factors of breastfeeding duration in preterm twin infants have a conflicting result. The purpose of this observational study was to compare feeding practices, at hospital discharge, of twin and singleton very low birth weight infants. The study is part of a prospective survey of a national Spanish cohort of very low birth weight infants (SEN1500) that includes 62 neonatal units. The study population comprised all infants registered in the network from 2002 to 2013. They were grouped into singletons and multiples. The explanatory variables were first analyzed using univariate models; subsequently, significant variables were analyzed simultaneously in a multiple stepwise backward model. During the twelve-year period, 32,770 very low birth weight infants were included in the database, of which 26.957 were discharged alive and included in this analysis. Nine thousand seven hundred and fifty-eight neonates were multiples, and 17,199 were singletons. At discharge, 31% of singleton infants were being exclusively breastfed, 43% were bottle-fed, and 26% were fed a combination of both. In comparison, at discharge, only 24% of multiple infants were exclusively breastfed, 43% were bottle-fed, and 33% were fed a combination of both (*p* < 0.001). On multivariable analysis, twin pregnancy had a statistically significant, but small effect, on cessation of breastfeeding before discharge (OR 1.10; 95% CI: 1.02, 1.19). Risks of early in-hospital breastfeeding cessation were also independently associated with multiple mother-infant stress factors, such as sepsis, intraventricular hemorrhage, retinopathy, necrotizing enterocolitis, intubation, and use of inotropes. Instead, antibiotic treatment at delivery, In vitro fertilization and prenatal steroids were associated with a decreased risk for shorter in-hospital breastfeeding duration. Multiple pregnancy, even in the absence of pathological conditions associated to very low birth weight twin infants, may be an impeding factor for in-hospital breastfeeding.

## 1. Introduction

Historically, formula was provided to very low birth weight (VLBW) infants to ensure rapid growth. However, recent studies show that feeding fortified human milk to preterm infants does not result in lower weight at discharge [1]. Achieving optimal development for preterm infants, whilst minimizing the risk for serious disease in the neonate, remains an important challenge for clinicians. Human milk is recommended for enteral feeding of preterm newborns [2], since it is associated with low rates of necrotizing enterocolitis (NEC) [3], sepsis, retinopathy of prematurity (ROP) [4], and a better neurocognitive development [5]. Despite efforts to offer human milk as the primary source of enteral nutrition during the stay in the neonatal intensive care unit (NICU), infant milk formulas are used in situations in which the mother’s milk or donor’s milk is not available.

Previous research has identified associations between the provision of breast milk in the NICU and maternal characteristics, infant characteristics, and environmental factors [6,7,8]. However, studies on predictive factors of breastfeeding duration in premature infants have conflicting results. Some studies have found that higher gestational age is associated with higher rates of breastfeeding [7,9], while others have found that mothers of VLBW infants acknowledge that human milk is a unique factor in improving outcomes for their infant [10] and, consequently, breastfeeding rates are higher among them.

The proportion of infants born preterm is higher for twins, when compared to singletons [11]. Breastfeeding poses unique challenges for the mothers of twins, but little attention has been paid to risk factors for early cessation of breastfeeding for these mothers. Only a few studies have focused on issues related to breastfeeding cessation for full term twins [12,13]. From a physiological standpoint, the World Health Organization (WHO) and other authorities state that mother’s milk is sufficient for feeding multiple babies [14]. It provides optimal nutrition for full term or preterm twins. A study comparing singleton and multiple premature, under 30 weeks, gestations, found that the milk volume was higher in mothers of multiples compared with singletons (599 mL/day vs. 430 mL/day) [15]. It has also been reported that the milk yield of individual breasts of mothers of twins is 0.8–2.1 kg per 24 h at six months, whereas the total milk yield of mothers of triplets can reach 3.0 kg per 24 h at 2.5 months [16]. Previous research shows that preterm twin infants may be less likely to be breastfed compared to singleton infants, but the data on this subject remains contradictory [17]. When interpreting the results from these studies, it should be noted that the type of babies included is variable, and that most studies do not provide details on feeding maintenance and complications. One issue that further complicates efforts to address this question is the finding that most preterm complications (NEC, ROP, sepsis, etc.) may be a cause of breastfeeding cessation, and may also be a consequence of breastfeeding cessation. This makes it difficult to explain the relationship between feeding type and severe clinical complications. The aim of this study is to shed light on the influence of major neonatal morbidity, and multiple birth, on in-hospital breastfeeding success. Specifically, we hypothesize that twin birth exerts its own substantial influence on poor breastfeeding outcomes among VLBW infants, irrespective of the clinical situations of these patients.

To test this hypothesis, we have used a multicenter, electronic health record-derived database that represents roughly two-thirds of all Spanish VLBW infants. The objective is to compare breastfeeding rates at hospital discharge of singleton and twin VLBW infants, and to evaluate which neonatal conditions and diseases preclude the provision of breast milk at the time of hospital discharge.

## 2. Methods

### 2.1. Design

The study is part of a prospective survey of a national Spanish cohort of VLBW infants (SEN1500 database), based on questionnaires. The SEN1500 network prospectively collects maternal and neonatal data of live-born infants of ≤1500 g who were born in or admitted to the collaborating NICUs within the first 28 days of life. Data were collected using a pre-established form and were submitted electronically using common and specific software. The characteristics, quality control, and data confidentiality systems of this database have been described elsewhere [18]. For the purpose of the present study, data corresponding to all discharged alive newborns within the database over a period of twelve years were retrieved. The SEN1500 database includes 62 neonatal units, which encompasses about two-thirds of all Spanish VLBW infants. The study population comprises all infants registered in the database, during the period 2002–2013. They were grouped into singletons and multiples. The same cohort was used previously to analyze morbidity and mortality associated with multiple birth [19].

### 2.2. Data Collection

In this study, we assess clinical factors that are known to influence breastfeeding during the hospital stay of singletons and twins. The main variable of interest was any breastfeeding, i.e., not being completely weaned from breastmilk, on the day when hospital discharge was completed. Data on maternal and infant characteristics were obtained from medical records by local health care professionals, using a pretested standardized questionnaire. The study variables were defined according to the variables catalogue of the SEN1500 database. We collected demographic data (birth weight, gender), obstetrical data (fertilization, antenatal care, gestational age, type of delivery, twinning), procedures and treatment (intra partum antibiotics, inotropic therapy, intubation, surfactant administration, ventilation, oxygen therapy, ductus closure, retinopathy surgery), neonatal morbidity (necrotizing enterocolitis, intraventricular haemorrhage, sepsis), length of hospitalization and type of feeding at discharge.

### 2.3. Outcomes

Exclusive breast milk feeding was defined as the infant receiving no food or drink other than the mother’s own milk. Any breast milk feeding was defined as the infant receiving mother’s own milk, independent of the addition of formula or other food and/or drink. SEN1500 was set up by the Spanish Association of Neonatologists. It encompasses 62 neonatal units, about two-thirds of all national neonatal units. The operational definitions used to collect feeding data at discharge are: No breastfeeding, any breastfeeding, or exclusive breastfeeding. The database systems are locally controllable. Breastfeeding outcomes are recorded after discharge, in each neonatal unit by the neonatologist responsible for registering obstetric and neonatal information, routinely collected from the Perinatal Health files.

### 2.4. Statistical Analysis

Data analysis was performed using statistical package SPSS 14.0 for Windows (IBM Company, Chicago, IL, USA). *p*-Values below 0.05 were considered to be of statistical significance. Descriptive statistics were used to outline mother and infant characteristics. Variables following binomial distributions were expressed as frequencies and percentages; and quantitative variables as median and ranges. Comparisons between qualitative variables were made using the Fisher Exact Test or Chi-square. Comparisons between quantitative and qualitative variables were performed through non-parametric tests (U of Mann-Whitney or Kruskal-Wallis).

Logistic regression (both univariate and multivariate) was used to assess the relationship between potential predictor variables of breastfeeding at discharge, a broad range of explanatory variables with a *p*-value under 0.1 was chosen to appear in the final regression model, yet this less restrictive approach is accepted, as well as considering explanatory variables one at a time [20]. Out of the variables with a significant relation to breastfeeding at discharge after the univariate analysis, we excluded those that, from a clinical point of view, are a result rather than a cause of feeding practice (anthropometric data at discharge). Finally, the explanatory variables included in the stepwise backward regression model were birth weight, height at birth, birth head circumference, gender, In vitro fertilization, multiple birth, prenatal care, prenatal administration of corticosteroids or antibiotics, delivery type, NICU use of surfactant or inotropic drugs, intubation, ventilation duration, oxygen therapy duration, CRIB score, duct pharmacological closure, surgical management of ROP, NEC, late onset sepsis (LOS), intraventricular hemorrhage (IVH), and supplementary oxygen at 36 weeks PMA (bronchopulmonary dysplasia). The results are presented as odds ratio (OR) with a 95% confidence interval (CI).

### 2.5. Statement of Ethics

The study was conducted in accordance with the Declaration of Helsinki. The Local Health District Human Research Ethics Committees approved the collection of the data from the SEN1500 database and subsequent analysis. Data collection in SEN1500 is anonymous and has been previously approved by the Ethics and Investigation Committees of the Collaborative Centers. It is in compliance with the Spanish data protection law. Ethical approval code: IIBSP-LEC-2011-129. Informed consent was obtained a legally authorized person, such as a parent or guardian.

## 3. Results

### 3.1. Characteristics of the Study Population

The study included 26,957 VLBW infants, of which 9758 were multiples, and 17,199 were singletons—who were all alive when discharged from the NICU. The characteristics of the study sample are summarized in Table 1. The median gestational age was 29 weeks, with a median birth weight of 1190 g. Female infants made up 50% of the sample. Cesarean delivery was common (72%), and 6% of infants were outborn. Use of antenatal care was observed in 22,830 (88%) patients. Antenatal corticosteroid administration rate was 84%, and median length of hospital stay was 50 days. As we have reported elsewhere [18], most patient characteristics and major morbidities differed between multiples and singletons (see Table 2).

### 3.2. Breastfeeding Patterns by Study Factors

At hospital discharge, 15,365 (57%) infants were being breastfed, 28% exclusively, and 29% non-exclusively. Out of the singleton infants, at hospital discharge, 31% were being exclusively breastfed, 43% exclusively bottle-fed, and 26% were being fed a combination of both. In comparison, at hospital discharge, only 24% of multiple infants were being exclusively breastfed, 43% exclusively bottle-fed, and 33% were being fed a combination of both. The distribution of the patients by feeding type at discharge differed in these two groups (*p* < 0.001).

### 3.3. Determinants of Breastfeeding at Hospital Discharge

The results of the univariate analysis are summarised in Table 3. With regards to mother and delivery characteristics, the unadjusted analysis shows that In vitro fertilization (IVF), prenatal care, administration of antenatal corticosteroids, intra partum antibiotics, and vaginal delivery, were all protective factors for being breastfed at discharge. Infant weight, height and head circumference at birth were also positively associated with breastfeeding rates at discharge. Additionally, infants that were being breastfed at discharge received less surfactant and were less often intubated or put on mechanical ventilation. They spent fewer days using supplemental oxygen, they received fewer inotropes and required less surgical treatment for ROP, and IVH (grades III and IV). NEC and LOS were more prevalent in infants not being breastfed at discharge.

However, when these factors were put into a multivariable model to determine the effect of a factor, while holding all others constant (Table 4), multiple birth (OR 1.10), surfactant administration (OR 1.10), prenatal care (OR 1.11), intubation (OR 1.15), LOS (OR 1.23), surgical treatment for ROP (OR 1.33), use of inotropes (OR 1.38), IVH grades III or IV (OR 1.42), bronchopulmonary dysplasia (OR 1.43) and NEC (OR 1.47), were significantly associated with a lower chance of any breastfeeding at discharge. On the other hand, the primary findings of this study indicate that the odds of a VLBW infant of being breastfed at the time of hospital discharge correlate to (a) the use of ibuprofen for ductus closure, (b) prenatal steroid treatment, (c) In vitro fertilization, and (d) intrapartum antibiotics.

## 4. Discussion

The incidence of multiple births has risen since the 1970s [21]. Within this group, the proportion of infants born preterm is higher compared to singletons [22]. There is strong evidence that not being fed breastmilk carries greater risks, with preterm babies having specific additional risks. However, mothers of preterm infants may terminate lactation earlier than mothers of full term infants [23]. Researching the obstacles for breastfeeding in preterm babies is a central issue in perinatology [8]. The low prevalence of breastfeeding among preterm infants may be explained by several factors. First, by difficulties inherent to their higher neonatal morbidity [24,25]. Also, maternal socio-demographic factors, and health-care services related factors may play an important role in the initiation and duration of breastfeeding. Although previous studies have examined a wide range of factors related to breastfeeding during a hospital stay for preterm infants, only a few have focused on the effects of multiple births on breastfeeding rates in this population.

Moreover, the few studies that have taken into account this variable have produced controversial results. The results on this issue are quite diverse depending on the study population and on the time of investigation of the nutritional method. To the best of our knowledge, from 1997 to 2017, four papers have found that the combination of prematurity and twin birth results in lower rates and shorter duration of breastfeeding among preterm multiples. On the other hand, six papers have found no breastfeeding differences between singleton and twin preterm infants. A 2004 study assessed exclusive and non-exclusive breastfeeding at day 1, day 3, week 2, month 1, and month 6 postpartum, among 346 mothers of full term and preterm singleton and multiple babies. They found that at any time point, except week 2 and month 1, the proportion of preterm multiples who fed at least some breast milk was less than in other groups [26]. A 2007 telephone survey on 94 VLBW infants reported that mothers of preterm multiples were 16 times more likely to provide formula before week 12 postpartum than mothers who delivered a singleton [27]. A 2014 Danish cohort of preterm infants (24–36 weeks of gestation) has found that multiple birth is significantly associated with the later establishment of exclusive breastfeeding [28]. A 2016 study on a very large sample of very preterm infants shows that the odds for exclusive breastfeeding at discharge for singletons was significantly higher than for twins [29]. Similarly, we found that singleton preterm infants were more likely to be exclusively breastfed at discharge than multiple preterm infants (31% versus 24%). Our results are in line with the previous research on the prevalence of exclusive breastfeeding for VLBW infants at NICU discharge, ranging from 11.4 to 30.5% in various countries [10,17,30]. All studies on this topic report that rates of breastfeeding (exclusive or non-exclusive) decrease with time. Exclusive breastfeeding at NICU discharge is essential to help women in continuing breastfeeding for durations recommended by the WHO. Accordingly, Kaneko et al. reported that the prevalence of exclusive breastfeeding was 22.6% at NICU discharge, and it had dropped only to 15.7% by the start of complementary feeding. These values are slightly higher or lower than those of recent large Japanese or Italian cohorts, respectively [17,31].

On the other hand, six authors have reported that preterm twins can breastfeed as successfully as preterm singleton infants. A 1997 paper on 15 singleton and 18 twin preterm infants showed no differences in feeding method from birth to discharge [32]. More recently, a Danish paper has reported no differences in exclusive breastfeeding rates at discharge [33], an Israeli paper has reported no differences in breastfeeding beyond six months of age [34], and three other papers from Western countries have shown no differences in any breastfeeding rates at discharge from the NICU [6,35,36].

For the majority of mothers, breastfeeding evolves naturally. However, it is not so when mothers experience the unique stress associated with the NICU environment [37]. Most maternal and NICU characteristics that challenge the success of breastfeeding have been extensively studied. Conversely, scarce attention has been devoted to the influence on breastfeeding of a variety of stress factors that preterm and fragile infants face. These include aggressive procedures, life-threatening and damaging conditions, such as IVH or ROP, without forgetting expected feeding difficulties related to the infant’s clinical status at feeding time. Therefore, in this study, we chose to examine the link of common neonatal medical complications with the infant continuing to receive breastmilk upon discharge.

VLBW infants are at risk for a variety of medical complications, any of which further limits the mother-infant dyad progression to breastfeeding competence. However, according to our multivariable analysis, only intubation, NEC, LOS, IVH grades III or IV, surgical ROP, the use of inotropes and twin pregnancy were risk factors for breastfeeding cessation before NICU discharge.

Our findings cannot substantiate previous research in this field, yet we have found only one article that identified absence of history of NEC or LOS as significant predictors of continued provision of mother’s milk in VLBW infants through to day of life 30 [38], and we have found no published research that reports that lower rates of breastfeeding among multiple preterm infants may reflect their distinct neonatal morbidity.

Our analysis of a very large sample of babies has reduced the number of variables linked to in-hospital breastfeeding cessation to fifteen independent risk factors. Twin pregnancy is one such risk factor, but its magnitude (OR = 1.10) is small enough to explain contradictory results in previous research on this topic. It is expected that small samples with bigger margins of error cannot detect this effect.

Finally, our research confirms that surgical NEC, ROP, bronchopulmonary dysplasia and LOS are linked to no breastfeeding at discharge. It is likely that this is a matter of reverse causation, since it is well established that lack of breastfeeding is a risk factor for such conditions in preterm infants. Likewise, it is reasonable to speculate that prenatal care, including prenatal steroid treatment, boost successful breastfeeding rates. Conversely, surfactant administration, intubation, inotropes, and IVH have not been previously linked to breastfeeding difficulties, and are, therefore, worth further investigation. On the other hand, the same applies to *ductus* closure, In vitro fertilization or *intra partum* antibiotics.

### Limitations

The large sample study, the exclusion of recall bias concerning breastfeeding duration and the diverse patient population, are strengths of our study. We were able to show some variation in maternal breast milk provision between multiple and singleton preterm infants. While these findings are important in broadening our knowledge, they are not without limitations.

Breastfeeding was not the focus of the primary study. Consequently, the questions concerning breastfeeding were less detailed than those that would be used in a prospective study. An additional limitation of this study is the use of maternal breast milk feedings at discharge to represent mothers continued provision of breast milk. We were also unable to quantify the volume of maternal breast milk received.

## 5. Conclusions

Our study of a very large sample of VLBW infants, shows that singleton pregnancy predicts a discrete increase of in-hospital breastfeeding rates, when compared to twin pregnancy. This study encourages a closer look at breastfeeding in the NICU. We support that breastfeeding among VLBW twin infants at NICU discharge is feasible; however, identification of multiple risk factors for the cessation of breastfeeding reflects the complexity of this process. By understanding predictive maternal or infant factors associated with poor breastfeeding outcomes, mothers at risk can be identified for more successful interventions. In addition, we can work to come up with adequate interventions for twin VLBW infants so that, when compared with singleton VLBW infants, neonatal morbidities will no longer be factors associated with failed breastfeeding. It is possible that early interventions for a few serious complications linked to breastfeeding cessation could have minimized the other problems and made it possible for the women to continue. Interventions in the NICU should enable mothers to transition to breastfeeding as soon as possible to promote success until discharge and beyond.

Going forward, for NICUS to excel at feeding human milk, in-depth work remains to determine barriers to a mother’s ability to provide breast milk and to establish evidence-based lactation care.

## Figures and Tables

**Table 1 nutrients-11-02191-t001:** Characteristics of 26,957 VLBW infants discharged alive from 62 Spanish neonatal units in 2002–2013.

Characteristic	Number (%)
Maternal Factors	
In vitro fertilization (%)	4747 (19)
Prenatal care (%)	22,830 (88)
Antenatal corticosteroid therapy (%)	22,059 (84)
Intrapartum antibiotic chemoprophylaxis (%)	11,502 (47)
Between hospital transfers	1651 (6)
Caesarean section (%)	19,503 (72)
Multiple birth (%)	9758 (36)
Median gestational age at birth, weeks (range)	29 (20–41)
Infant Factors	
Male sex (%)	13,374 (50)
Apgar score at 1 min (%)	7 (0–10)
Median birth weight, g. (range)	1190 (360–1499)
No breastfeeding at discharge (%)	11,592 (43)
Morbidities	
Early onset sepsis (%)	950 (4)
Late onset sepsis (%)	7672 (29)
Necrotising enterocolitis (any) (%)	1506 (6)
Bronchopulmonary dysplasia (%)	3127 (14)
Surgical management of ROP (%)	951 (4)
Median length of stay, days (range)	50 (1–238)

Data are presented as median (minimum–maximum) or number (percentile), unless specified. Abbreviations: ROP, retinopathy of prematurity; VLBW, very low birth weight; g, gram.

**Table 2 nutrients-11-02191-t002:** Demographic and pathological characteristics of singletons versus multiples.

Variables	Singleton	Multiple	*p*-Value
	*n* = 17,199	*N* = 9758	
Maternal Factors			
Median gestational age (weeks) (range)	29 (20–41)	30 (20–39)	<0.001
In vitro fertilization	833 (5.2%)	3914 (44%)	<0.001
Outborn	1153 (6.7%)	498 (5.1%)	<0.001
Antenatal care	14,320 (86.3%)	8510 (90.8%)	<0.001
Antenatal steroids	13,741 (81.9%)	8318 (86.9%)	<0.001
Maternal *intra partum* antibiotics	7172 (46.3%)	4330 (49%)	<0.001
Vaginal delivery	5510 (32%)	1944 (19.9%)	<0.001
Infant Factors			
Median birthweight (g) (range)	1160 (360–1499)	1220 (395–1499)	<0.001
Male sex (%)	8728 (50.7)	4646 (47.6)	<0.001
Morbidities			
Days of ventilatory therapy	0 (0–835)	0 (0–672)	<0.001
Days of supplemental oxygen	4 (0–391)	3 (0–286)	<0.001
Surfactant at any time	7689 (45%)	4130 (42.6%)	<0.001
Inotropic therapy	3728 (22.7%)	1959 (21.1%)	0.003
Necrotizing enterocolitis (all grades)	1007 (5.9%)	499 (5.1%)	0.011
Early-onset sepsis	671 (3.9%)	279 (2.9%)	<0.001
Late-onset sepsis	5173 (30.5%)	2499 (26%)	<0.001
IVH (all grades)	3471 (21.7%)	1494 (16.5%)	<0.001
Supplementary Oxygen at discharge	1074 (6.3%)	507 (5.2%)	<0.001
NICU length of stay (days)	19 (0–187)	14 (0–238)	<0.001

Data are presented as mean (SD) or number (percentile), unless specified. Abbreviations: IVH, intraventricular hemorrhage; NICU, neonatal intensive care unit. Comparisons between qualitative variables were made using the Fisher Exact Test or Chi-square. Comparisons between quantitative and qualitative variables were performed through non-parametric tests (U of Mann-Whitney or Kruskal-Wallis).

**Table 3 nutrients-11-02191-t003:** Univariate analysis of risk factors for no breastfeeding at NICU discharge.

Characteristics	Twins		Single	
	OR (95% CI)	*p*	OR (95% CI)	*p*
Birth weight	1 (1–1)	0.008	0.99 (0.99–0.99)	<0.001
Height at birth	0.99 (0.99–1)	0.011	0.99 (0.99–0.99)	<0.001
Birth head circumference	0.99 (0.99–1)	0.1	0.99 (0.99–0.99)	<0.001
Male gender	1.08 (0.99–1.17)	0.076	0.99 (0.93–1.05)	0.65
Twin birth		---		---
In vitro fertilization	0.93 (0.85–1.01)	0.099	0.59 (0.5–0.68)	<0.001
Prenatal care	0.84 (0.72–0.97)	0.022	1.46 (1.33–1.6)	<0.001
Prenatal steroids:		<0.001		<0.001
No				
Partial	0.66 (0.57–0.77)		0.84 (0.76–0.93)	
Complete	0.72 (0.63–0.82)		0.74 (0.68–0.8)	
Antibiotic treatment at delivery	0.72 (0.66–0.79)	<0.001	0.86 (0.81–0.92)	<0.001
Delivery type: C-section	1.12 (1.01–1.24)	0.038	0.87 (0.82–0.93)	<0.001
Intubation:	1.22 (1.1–1.35)	<0.001	1.58 (1.48–1.69)	<0.001
CRIB score	1.06 (1.04–1.08)	<0.001	1.11 (1.09–1.12)	<0.001
Conventional ventilation duration	1.01 (1.01–1.02)	<0.001	0.58 (0.55–0.62)	<0.001
Oxygen therapy duration	1 (1–1.01)	<0.001	0.62 (0.55–0.69)	<0.001
Surfactant administration	1.15 (1.06–1.25)	0.001	1.56 (1.46–1.66)	<0.001
Inotropic therapy	1.53 (1.38–1.7)	<0.001	1.96 (1.82–2.12)	<0.001
*Ductus* closure:		0.022		<0.001
None				
Indometacin	1.12 (0.98–1.28)		1.51 (1.36–1.67)	
Ibuprofen	0.88 (0.77–1)		1.2 (1.08–1.32)	
ROP surgery	1.42 (1.13–1.78)	0.002	2.43 (2.06–2.88)	<0.001
Necrotizing enterocolitis	1.65 (1.37–1.99)	<0.001	1.88 (1.65–2.15)	<0.001
Late onset sepsis	1.27 (1.15–1.39)	<0.001	1.59 (1.48–1.7)	<0.001
Intraventricular hemorrhage		0.001		<0.001
No				
Grade I–II	1.13 (0.99–1.28)		1.24 (1.14–1.36)	
Grade III–IV	1.48 (1.18–1.85)		1.94 (1.67–2.26)	
Supplemental oxygen *at* 36 weeks *PMA*	1.53 (1.33–1.76)	<0.001	2 (1.82–2.21)	<0.001

Data presented as median (range) or percentile. ABBREVIATIONS: NICU, neonatal intensive care unit; PMA, postmenstrual age; ROP, retinopathy of prematurity.

**Table 4 nutrients-11-02191-t004:** Logistical regression analysis of no breastfeeding at discharge as a function of selected predictor variables.

Variables	Odds Ratio (95% CI)	*p*-Value
	Maternal Factors	
Prenatal care	1.11 (1.01–1.22)	0.036
Partial prenatal steroids	0.79 (0.71–0.88)	<0.001
Complete prenatal steroids	0.82 (0.75–0.89)	<0.001
In vitro fertilization	0.84 (0.77–0.92)	<0.001
Antibiotic treatment at delivery	0.86 (0.81–0.92)	<0.001
Multiple birth	1.10 (1.02–1.19)	0.009
	Infant Factors	
Surfactant administration	1.10 (1.01–1.19)	0.023
Endotracheal intubation	1.15 (1.06–1.25)	0.001
	Morbidities	
Ibuprofen treatment for ductus closure	0.73 (0.66–0.81)	<0.001
Late onset sepsis	1.27 (1.18–1.36)	<0.001
Surgical management of ROP	1.33 (1.13–1.57)	0.001
Inotrope support	1.38 (1.26–1.50)	<0.001
Grade III–IV intraventricular hemorrhage	1.42 (1.21–1.67)	<0.001
Supplemental oxygen at 36 weeks PMA	1.43 (1.3–1.59)	<0.001
Necrotizing enterocolitis	1.47 (1.29–1.69)	<0.001

−2 log likelihood: 23234; DF: 17. Abbreviations: ROP, retinopathy of prematurity.

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
