# Peer review of "Breastfeeding Disparities between Multiples and Singletons by NICU Discharge"

_nutrients, 2019, doi:10.3390/nu11092191_

Round 1

Reviewer 1 Report

Review:

Breastfeeding disparities between multiples and singletons by NICU discharge

This study compared the success of singleton and twin VLBW infants in maintaining breastfeeding during their hospital stay, and to evaluate which neonatal conditions and diseases preclude provision of breast milk at the time of hospital discharge using multicentre electronic health record-derived database, which represents roughly two-thirds of all Spanish VLBW infants.

The findings of this manuscript are important in promoting early breastfeeding initiation and hence improving health and survive of twin VLBW infants.

Specific comments:

INTRODUCTION:

“The objective is to compare the success of singleton and twin VLBW infants in maintaining breastfeeding during their hospital stay”: as the authors used a breastfeeding outcome only at discharge (not throughout the hospital stay), it is hard to understand how this objective could be achieved (see the limitation Line 237-238).

The authors need to provide their hypothesis related to the research topic.

neonatal unit (NICU)”: is not correct full phrase for NICU.

Methods

Line 69: Design

It is not clear how the study sample was chosen from the SEN1500 database.

based on questionnaires”: which questionnaires (reference?) the authors referred to? Were the questionnaires validated? The authors need to provide more details on the databse.

Line 76: Data collection

The author should give details of the main variable of interest and all explanatory variables (probably classified into groups) used in their analysis.

Line 83: Outcomes

These were definitions of exclusive and any breastfeeding. The authors need to describe how their breastfeeding outcome variables were collected in the SEN1500 database.

Line 87: Statistical analysis

variables with a p-value under 0.1 were analysed simultaneously in a multiple stepwise backward model”:  the authors need to explain with a reference why p <0.1 was used as the cut-off value for candidate explanatory variables to be included in their multiple stepwise backward model? In addition, the authors need to mention which regression analysis was used. Did the authors concern about any issue with multicollinearity when including many covariates in their multivariable model?

The authors should mention statistical tests used for comparing the difference in sample (Demographic and pathological) characteristics between singleton and twin VLBW infants staying in NICU. 

“The variables were broken down into two groups”: which two groups and the purpose of the two groups related to multiple stepwise backward modelling?

“…and other non-significant factors that did not influence the final model”: the authors need to indicate whether p<0.05 was used for the final model.

RESULTS

Line 110: Characteristics of the Study Population.

The study included 32,770 preterm infants, of which 11,647 were multiples and 21,123 were singletons”: were all these infants were VLBW and stayed in NICU? The authors need to define their study population clearly, i.e., whether in their study only VLBW singleton and twin infants staying NICU were included. In view of this point, the authors need to modify their paper title, given that in general not all singleton and twin infants are VLBW and stay in NICU.

Line 112 to 116: The authors need to carefully check their paragraph: some figures mentioned were not matched to figures in Table 1.

Line 119: Table 1

Data are presented as median (minimum‐maximum) or number (percentile)”: median should be presented with interquartile range.  

Based on the author’s objectives, this table should be stratified in columns by singleton and twin VLBW infants staying in NICU, and explanatory variables should listed in rows by groups (maternal, infants, delivery, pathological etc).

Line 122: Table 2:

This table should be combined with Table 1, see comments above.

For all figures, decimal separator should be a dot “.” rather a comma “,”.

The tests used for testing the difference between VLBW singleton and twin infants staying NICU should be mentioned in the Statistical analysis section and again as a footnote of this table. The authors are suggested to consult a statistician regarding their analyses, for example, the narrow 95% Cis caused by large sample size.

Line 125: Breastfeeding Patterns by Study Factors

“At hospital discharge, 15,037 (56%) infants were being breastfed”: this % is not correct given that “The study included 32,770 preterm infants”.

Determinants of Breastfeeding at Hospital Discharge

Table 3 is not an appropriate table to assess the crude association between these factors and breastfeeding. The authors should perform simple logistic regression analysis to report the crude odds ratio for each factor of interest; again this table should be stratified by VLBW singleton and twin infants staying NICU.

The authors should make a justification on why they assessed the difference in exclusive breastfeeding prevalence between VLBW singleton and twin infants staying NICU at discharge, however identified the determinants for any breastfeeding.

“multivariate” should be replaced by “Multiple” (or Multivariable”). Statistically “Multivariate” refers to multivariate analysis, which is not the case for the present study, and suggest correcting.

Line 154: Table 4

This table only displays the overall factors. It is suggestive that independent factors influencing breastfeeding at discharge should be presented for VLBW singleton and twin infants staying NICU separately for comparisons. Again this table should be stratified by VLBW singleton and twin infants staying NICU.

In the footnote of Table 4, the authors should attach the following information:

–2 log likelihood (deviance) and degree of freedom for the final model

All variables included in the initial model

the regression strategy used to obtained this final model

DISCUSSION:

Line 220-221:

Our analysis of a very large sample of babies has reduced the number of variables linked to in-220 hospital breastfeeding cessation to seven independent risk factors”:  Table 4 includes 11 factors.  

The authors should include their more discussions of other important factors presented in Table 4.

Conclusions:

The authors should indicate how their findings is going to add or change existing practices in promoting early breastfeeding (at discharge or throughout the hospital stay) to improve the breastfeeding rate among mothers of twin VLBW infants staying in NICU in Spain.

Author Response

INTRODUCTION:

The objective is to compare the success of singleton and twin VLBW infants in maintaining breastfeeding during their hospital stay”: as the authors used a breastfeeding outcome only at discharge (not throughout the hospital stay), it is hard to understand how this objective could be achieved (see the limitation Line 237-238).

ANSWER: We have changed the aim to: “The objective is to compare breastfeeding rates at hospital discharge of singleton and twin VLBW infants” (Line 93).

The authors need to provide their hypothesis related to the research topic.

ANSWER: We hypothesize that twin pregnancy exert its own substantial influence on poor breastfeeding outcomes (Line 89-90).

neonatal unit (NICU)”: is not correct full phrase for NICU.

ANSWER: Done (Line 59).

Methods

Line 69: Design

It is not clear how the study sample was chosen from the SEN1500 database.

ANSWER: The sample comprised all infants registered in the database, during the period 2002-2013. (Line 107, 108)

based on questionnaires”: which questionnaires (reference?) the authors referred to? Were the questionnaires validated? The authors need to provide more details on the database.

ANSWER: The Spanish SEN1500 network prospectively collects maternal and neonatal data of live-born infants of ≤1,500 g born in or admitted to the collaborating NICUs within the first 28 days of life. Data were collected using a pre-established form and were submitted electronically using specific common software. The characteristics, quality control, and data confidentiality systems of this database have been described elsewhere (Line 99-106) [18, new reference added to bibliography].

Line 76: Data collection

The author should give details of the main variable of interest and all explanatory variables (probably classified into groups) used in their analysis.

ANSWER: The main variable of interest was any breastfeeding, i.e. not being completely weaned from breastmilk, on the day when hospital discharge was completed. (Line 112,113)

We collected demographic data (birthweight, gender), obstetrical data (fertilization, antenatal care, gestational age, type of delivery, twinning), procedures and treatment (intra partum antibiotics, inotropic drugs use, intubation, surfactant treatment, ventilation, oxygen therapy, ductus closure, retinopathy surgery), neonatal morbidity (necrotizing enterocolitis, intraventricular haemorrhage, sepsis), length of hospitalization and type of feeding at discharge. (Line 116-121)

Line 83: Outcomes

These were definitions of exclusive and any breastfeeding. The authors need to describe how their breastfeeding outcome variables were collected in the SEN1500 database.

ANSWER: SEN1500 was set up by the Spanish Association of Neonatologists. It encompasses 62 neonatal units, about two thirds of all national neonatal units. The operational definitions used to collect infant feeding data at discharge are: no breastfeeding, any breastfeeding or exclusive breastfeeding. The database systems are locally controllable. Breastfeeding outcomes are recorded after discharge, in each neonatal unit, by the neonatologist responsible for registering obstetric and neonatal information routinely collected from the Perinatal Health files. (Line 125-131)

Line 87: Statistical analysis

variables with a p-value under 0.1 were analysed simultaneously in a multiple stepwise backward model”:  the authors need to explain with a reference why p <0.1 was used as the cut-off value for candidate explanatory variables to be included in their multiple stepwise backward model?

ANSWER: A broad range of explanatory variables with a p-value <=0.1 was chosen to appear in the final regression model, yet this less restrictive approach is accepted as well as considering explanatory variables one at a time. (new reference: 20) (Line 141-144)

In addition, the authors need to mention which regression analysis was used. Did the authors concern about any issue with multicollinearity when including many covariates in their multivariable model?

 ANSWER: Multiple logistic regression was used to assess the relationship between more than two continuous or categorical explanatory variables and a single categorical response variable (breastfeeding at discharge).

Multicollinearity, the fact that some predictor variables are correlated among themselves does not tend to affect inferences about mean responses and the goodness-of-fit statistics (Applied Linear Statistical Models, p289, 4th Edition). As our primary goal is to make predictions, we don’t need to reduce moderate multicollinearity.

The authors should mention statistical tests used for comparing the difference in sample (Demographic and pathological) characteristics between singleton and twin VLBW infants staying in NICU. 

ANSWER: a paragraph has been added to Statistical Analysis (Line 136-139)-

The variables were broken down into two groups”: which two groups and the purpose of the two groups related to multiple stepwise backward modelling?

ANSWER: We apologyze, the sentence is redundant and confusing, it has been deleted (Line 151).

“…and other non-significant factors that did not influence the final model”: the authors need to indicate whether p<0.05 was used for the final model.

ANSWER: Done (Line 155).

Reviewer 2 Report

The authors analyzed a large cohort of 32,770 infants below 1,500g birth weight admitted to 62 NICUs in Spain 2002-2013. Mortality - one of the most important variables - is missing, and some babies were hospitalized for just one day because they died. Thus, it is impossible to use "feeding at discharge" as the primary outcome variable for all babies in the same fashion. There are two ways to deal with the problem:

Kaplan-Meier analysis with a Cox proportional hazard model to assess the impact of input variables

Restriction of the analysis to infants who were discharged home alive.

The results are likely to be quite different.

There are some further flaws in the statistical analysis:

Variables such as gestational age and birth weight will not be normally distributed in a cohort of infants defined by an upper birth weight of 1500 g. Therefore, both descriptive and analytical statistics should be based on non-parametric tests (e.g. replace mean and SD by median and range)

OR can only be calculated for categorical variables, not for continuous ones

Delineate decimals by a dot, not by a comma 

Author Response

The authors analyzed a large cohort of 32,770 infants below 1,500g birth weight admitted to 62 NICUs in Spain 2002-2013. Mortality - one of the most important variables - is missing, and some babies were hospitalized for just one day because they died. Thus, it is impossible to use "feeding at discharge" as the primary outcome variable for all babies in the same fashion. There are two ways to deal with the problem:

Kaplan-Meier analysis with a Cox proportional hazard model to assess the impact of input variables

Restriction of the analysis to infants who were discharged home alive.

ANSWER: we have restricted the analysis to infants who were discharged alive

The results are likely to be quite different.

ANSWER: Patient characteristics have changed, the main results are not quite different. We have added variables to the multivariate analysis

There are some further flaws in the statistical analysis:

Variables such as gestational age and birth weight will not be normally distributed in a cohort of infants defined by an upper birth weight of 1500 g. Therefore, both descriptive and analytical statistics should be based on non-parametric tests (e.g. replace mean and SD by median and range)

DONE

OR can only be calculated for categorical variables, not for continuous ones

DONE (TABLES 1, 2 & 3)

Delineate decimals by a dot, not by a comma 

DONE

Round 2

Reviewer 1 Report

Thanks for the authors’ efforts on addressing the reviewer’s comments. In this revised manuscript, the authors have improved their Introduction and Methods sections according to the reviewer’s comments.

Unfortunately, the authors did not respond reviewer’s feedback/comments on their Results, Discussion and Conclusions sections (see Report 1) in their response file ‘coverletter 1’, and no changes were made addressing the reviewer’s comments on these sections in their revised paper.

The contents of the paper (Discussion and Conclusions sections) need to be enriched and specified, and the presentations of results (all tables) need to be improved.

Author Response

We would like to thank Reviewer 1 for his/her time spent on reviewing our manuscript and for his/her thoughtful comments and efforts towards improving our study. We have taken the comments on board to clarify the manuscript. Please find below a detailed point-by-point response to all comments. Our answers are in italics.

We apologise for our unintentional mistake, we overlooked Reviewer’s 1 comments on Results, Discussion and Conclusions in our first amendment.

Since the restructuring of the manuscript was substantial, Line numbering refers to the revised manuscript, attached as a supplement to the Editor Response

RESULTS

Line 110: Characteristics of the Study Population.

The study included 32,770 preterm infants, of which 11,647 were multiples and 21,123 were singletons”: were all these infants were VLBW and stayed in NICU? The authors need to define their study population clearly, i.e., whether in their study only VLBW singleton and twin infants staying NICU were included. In view of this point, the authors need to modify their paper title, given that in general not all singleton and twin infants are VLBW and stay in NICU.

ANSWER:

We have followed Reviewer's 2 advice about exclusion criteria. Therefore, the new sample includes exclusively VLBW infants that were discharged alive from the NICU. Abstract: Line 36-39; Design: Line 110-118; Results: Line 188-189.

Line 112 to 116: The authors need to carefully check their paragraph: some figures mentioned were not matched to figures in Table 1.

ANSWER: The sample and the figures have changed, we have checked that figures on Line 188-196 match to figures on Tables.

Line 119: Table 1

Data are presented as median (minimum-maximum) or number (percentile)”: median should be presented with interquartile range.

ANSWER

The minimum and the maximum values, the range, and the interquartile range are commonly used measures of dispersion. There is no strict rule to select one of them. We present minimum-maximum because it can be used as a measure of variability in large samples (N>10,000) without too many outliers. In spite of several limitations, minimum-maximum gives a quick and easy way to estimate the spread of data.

Based on the author’s objectives, this table should be stratified in columns by singleton and twin VLBW infants staying in NICU, and explanatory variables should listed in rows by groups (maternal, infants, delivery, pathological, etc.).

ANSWER: All tables include only VLBW infants alive until NICU discharge. This table has been stratified as requested.

Line 122: Table 2:

This table should be combined with Table 1, see comments above.

ANSWER:

We do not completely agree to combine Table 1 and Table 2. We think it is more informative to devote Table 1 to overall baseline characteristics of study participants. Some authors present their results in this way (Mikami 2018, among many others). Conversely, other authors (Zachariassen 2010, among many others) combine patients' characteristics, univariate analysis and final models in the same table. Any approach has its limitations. It is not difficult to pay attention to a table with two columns, it gives an overall picture of the sample. Tables with more than four columns are ideal to compare different groups within the same sample.

Mikami FCF et al. Breastfeeding Twins: Factors Related to Weaning. J Hum Lact. 2018

Nov;34(4):749-759. doi: 10.1177/0890334418767382.

Zachariassen G et al. Factors associated with successful establishment of breastfeeding in very preterm infants. Acta Paediatr. 2010 Jul;99(7):1000-4. doi:10.1111/j.1651-2227.2010.01721.x.

For all figures, decimal separator should be a dot “.” rather a comma “,”.

ANSWER: done.

The tests used for testing the difference between VLBW singleton and twin infants staying NICU should be mentioned in the Statistical analysis section and again as a footnote of this table. The authors are suggested to consult a statistician regarding their analyses, for example, the narrow 95% Cis caused by large sample size.

ANSWER: The tests used for testing the difference between VLBW singleton and twin infants discharged alive from the NICUs are now mentioned in the Statistical analysis section (Line 155-157) and again as a footnote of Table 2.

Smaller sample sizes would generate wider confidence intervals. There is an inverse square root relationship between confidence intervals and sample sizes.

Line 125: Breastfeeding Patterns by Study Factors

“At hospital discharge, 15,037 (56%) infants were being breastfed”: this % is not correct given that “The study included 32,770 preterm infants”.

ANSWER: Sorry about our previous mistake. The sample and the figures have changed (LINE 209-215).

Determinants of Breastfeeding at Hospital Discharge

Table 3 is not an appropriate table to assess the crude association between these factors and breastfeeding. The authors should perform simple logistic regression analysis to report the crude odds ratio for each factor of interest; again this table should be stratified by VLBW singleton and twin infants staying NICU.

ANSWER: Table 3 refers exclusively to VLBW infants (both twins and singletons) who were discharged alive from the NICU. Table 3 was obtained performing a simple logistic regression. We have added crude ORs and 95%CI to table 3 as required by Reviewer 1.

The authors should make a justification on why they assessed the difference in exclusive breastfeeding prevalence between VLBW singleton and twin infants staying NICU at discharge, however identified the determinants for any breastfeeding.

ANSWER: Any breast milk feeding was defined as the infant receiving mother’s own milk, independent of the addition of formula or other food and/or drink (Outcomes, LINE: 138-141). Therefore, any breast-feeding excludes only no breastfeeding or exclusive formula-feeding, what is the same. Tables 3 & 4 refer to risk factors for no breastfeeding at NICU discharge. Our multivariable approach analyses risk factors for no breastfeeding at discharge.

We have also assessed the distribution of patients by feeding type (exclusively breastfed, exclusively bottle-fed, or fed a combination of both) at discharge in two groups of patients: multiples and singletons VLBW infants (LINE 209-215).

“multivariate” should be replaced by “Multiple” (or Multivariable”). Statistically “Multivariate” refers to multivariate analysis, which is not the case for the present study, and suggest correcting.

ANSWER: Done, LINE: 43,232,311.

Line 154: Table 4

This table only displays the overall factors. It is suggestive that independent factors influencing breastfeeding at discharge should be presented for VLBW singleton and twin infants staying NICU separately for comparisons. Again this table should be stratified by VLBW singleton and twin infants staying NICU.

ANSWER: This table lists all factors independently associated with the absence of breastfeeding at discharge from neonatal units in VLBW infants. The table presents ORs and 95% CIs for no breast feeding in adjusted analysis. The title of this table mentions the main outcome (no breastfeeding). General concepts embodied in the table: key predictor (twin pregnancy) and confounders. Multivariable data is often of huge size that will most likely result a dense structure. Hence, one difficulty with multivariable data is their visualization. We agree that there are different ways to report multivariable analysis results, but we highlight that tables similar to table 4 belong to the standard way of visualizing multivariable analyses data (Ericson 2016, among other authors).

Ericson J. et al. Changes in the prevalence of breastfeeding in preterm infants discharged from neonatal units: a register study over 10 years. BMJ Open. 2016 Dec 13;6(12):e012900. doi:10.1136/bmjopen-2016-012900.

In the footnote of Table 4, the authors should attach the following information:

–2 log likelihood (deviance) and degree of freedom for the final model

All variables included in the initial model

the regression strategy used to obtained this final model

ANSWER: -2 log likelihood ratio and degrees of freedom are now included in the footnote of Table 4. The deviance (-2LL) depends on the sample size as well as in the number of parameters in the model. Our research on neonatal morbidity and breastfeeding does not deal with well known educational and social predictors of early breastfeeding cessation. The regression strategy is described in the section of Statistical Analysis.

DISCUSSION:

Line 220-221:

Our analysis of a very large sample of babies has reduced the number of variables linked to in-220 hospital breastfeeding cessation to seven independent risk factors”: Table 4 includes 11 factors.

The authors should include their more discussions of other important factors presented in Table 4.

ANSWER: As the sample has changed, the results of the analysis have changed. Table 4 includes now 15 factors. They are considered in the Discussion (LINE 327-335).

Conclusions:

The authors should indicate how their findings is going to add or change existing practices in promoting early breastfeeding (at discharge or throughout the hospital stay) to improve the breastfeeding rate among mothers of twin VLBW infants staying in NICU in Spain.

ANSWER: a paragraph on in-depth studies and interventions to support NICUS to excel in breast milk feeding and to establish evidence based lactation care has been added to the Conclusions (LINE 350-365).

Reviewer 2 Report

The authors have changed the Results section to give the data of the infants discharged alive. However, the Abstract still contains the data of all infants admitted, and the table gives data of infants discharged alive with a legend saying "infants admitted". The authors should provide the number of all infants admitted (32.770) and the infants discharged alive (26.957) and then go on just with the infants discharged alive. The table also should list the duration of hospitalization. 

Author Response

We thank very much the comments of Reviewer 2.

We forgot to correct the figures of the Abstract, we apologyze for this.

ANSWERS:

The Abstract has been modified, we provide the number of all infants admitted (32,770) and the number of all infants discharged alive (26,957). The title of Table 1 has been modified to:

Characteristics of 26,957 VLBW infants discharged alive from 62 Spanish neonatal units in 2002–2013.

Table 2 includes NICU length of stay for twins and singletons.
